# Deep Sensing for Compressive Video Acquisition [note 1]

**DOI:** 10.3390/s23177535

**Published:** 2023-08-30

**Authors:** Michitaka Yoshida, Akihiko Torii, Masatoshi Okutomi, Rin-ichiro Taniguchi, Hajime Nagahara, Yasushi Yagi

**Affiliations:** 1Japan Society for the Promotion of Science, Shizuoka University, Hamamatsu 102-0083, Japan; 2Department of Systems and Control Engineering, School of Engineering, Tokyo Institute of Technology, Tokyo 152-8550, Japan; 3Faculty of Information Science and Electrical Engineering, Kyushu University, Fukuoka 819-0395, Japan; 4Institute of Datability Science, Osaka University, Suita 565-0871, Japan

**Keywords:** deep sensing, deep optics, compressive sensing, video reconstruction, deep neural network

## Abstract

A camera captures multidimensional information of the real world by convolving it into two dimensions using a sensing matrix. The original multidimensional information is then reconstructed from captured images. Traditionally, multidimensional information has been captured by uniform sampling, but by optimizing the sensing matrix, we can capture images more efficiently and reconstruct multidimensional information with high quality. Although compressive video sensing requires random sampling as a theoretical optimum, when designing the sensing matrix in practice, there are many hardware limitations (such as exposure and color filter patterns). Existing studies have found random sampling is not always the best solution for compressive sensing because the optimal sampling pattern is related to the scene context, and it is hard to manually design a sampling pattern and reconstruction algorithm. In this paper, we propose an end-to-end learning approach that jointly optimizes the sampling pattern as well as the reconstruction decoder. We applied this deep sensing approach to the video compressive sensing problem. We modeled the spatio–temporal sampling and color filter pattern using a convolutional neural network constrained by hardware limitations during network training. We demonstrated that the proposed method performs better than the manually designed method in gray-scale video and color video acquisitions.

## 1. Introduction

Digital cameras use light to capture information about a scene in the real world. However, image sensors usually consist of photodiodes arranged in two dimensions, and it is not possible to directly capture the scene information, because the information about the scene has more dimensions, such as space, time, and color. Therefore, it is necessary to map this high-dimensional information into two dimensions using a sensing matrix. Formally, image capture using a sensing matrix is expressed as follows:(1)$$y\left(w,h\right)=\sum_{c,t}^{C,T}\phi \left(w,h,c,t\right)⊙x\left(w,h,c,t\right),$$
where $y\in R^{W\times H}$ is the two-dimensional (2D) captured image, $\phi \in R^{W\times H\times C\times T}$ is the sensing matrix, ⊙ is the element-wise product, and $x\in R^{W\times H\times C\times T}$ is the target scene (*W*: width of image, *H*: height of image, *C*: number of channels, and *T*: number of frames). The color and time information of x often are mapped to the neighbors of the pixel y using the sensing matrix ϕ. The target scene x is reconstructed from captured image y, but the captured image is less dimensional than the scene, making it an ill-posed problem. Traditional methods reconstruct the target scene by interpolation [1,2], sparse reconstruction [3,4,5,6,7], or a Gaussian mixture model (GMM) [8]; or, recently, deep neural network (DNN)-based models [9,10,11,12,13,14,15] have been used for higher speed and quality. Here, the sensing matrix and the reconstruction method are related. The sensing matrix can be described as the convolution of scene information into the captured image and the reconstruction as the deconvolution from the captured image. The task of convolving the data into features and recovering the original data by deconvolution is the same as that of an autoencoder [16]. Therefore, methods have been proposed to jointly optimize the sensing matrix and reconstruction using deep learning, as in an autoencoder [9,10,15,17,18,19,20,21]. This approach is called deep sensing, and it obtains better results than optimizing sensing and reconstruction separately. We adopt the deep sensing approach to color video sensing. While previous methods using deep sensing optimize the measurement matrix in one dimension, our proposed method optimizes the observation matrix in the color and temporal dimensions with reconstruction.

The color information is usually obtained by distributing it to neighboring pixels using a uniform color filter array (CFA). The Bayer filter [1] is often used for this color filter, and it consists of red (R), green (G), and blue (B) filters (as shown in Figure 1a). Because the RGB filter cuts out a lot of light, filters such as RGB-white (W) and cyan (C), magenta (M), and yellow (Y) filters have also been proposed to increase sensitivity [1,22,23]. When a uniform CFA is used, the spatial resolution of each color is decreased to 1/c (*c* is the number of color channels). The reconstruction of color information from uniformly sampled images is called demosaicing. The existing demosaicing methods are just improvements of linear interpolation, and the reconstruction quality is limited because of the lack of captured information [24]. Therefore, compressive sensing has been used to obtain color information using a random CFA [2,25] (Figure 1b). The compressive sensing theory assumes that information is represented by a small number of coefficients (or a sparse representation vector) on a given basis. Moreover, the original signal can be recovered from a small number of observations. If the target scene is unknown, a random CFA is optimal, but if the target scene is known, there exists an optimal CFA that is better than a random one [9,10,26]. Furthermore, because a CFA is a matrix that selects the channel, it is easy to represent it in a DNN, and it is hence possible to optimize the CFA using a DNN. This approach is able to jointly optimize CFA and reconstruction using an end-to-end DNN because sensing can be represented using convolution, and reconstruction can be represented by deconvolution [9,10]. Methods for compressing and acquiring spatio–temporal information can be divided into the temporal compression method and the spatial compression method [27]. The spatial compression method [28] is similar to a single-pixel camera [29]. It acquires an image as a single observation by using a high-speed encoding sensor. On the other hand, the temporal compression method compresses spatio–temporal information and acquires the video with a single image. Traditionally, it captured spatio–temporal information by shifting the exposure times during the capture of a single frame in the neighboring pixels and sacrificing spatial resolution [30,31] (Figure 1c). This capturing method reduces the spatial resolution to 1/T (*T*: number of frames). For example, the first sub-pixel is exposed at the first sub-frame, the second sub-pixel is exposed at the second sub-frame, and the last sub-pixel is exposed at the last sub-frame. Although it is possible to recover the original resolution using a super-resolution method, such methods are only an estimation using the few data captured by uniform sampling, and they do not always reconstruct the original scene. Gupta et al. [31] chose between capturing space or time information for each region but were not able to overcome the trade-off between the spatial and temporal dimensions. In contrast, compressed sensing can be used to acquire spatial and temporal information simultaneously without reducing spatial resolution or color information. By capturing coded images using spatio–temporal random exposure, the spatio–temporal information is convoluted into a single image, and the video is reconstructed in post-processing (Figure 1d). Before the appearance of deep learning, videos were reconstructed by iterative optimization such as the L0 norm minimization problem [3,4], the L1 norm minimization problem [5,6,7], a framework that combines motion estimation and the Kalman filter [32], and Gaussian mixture model-based methods [8]. In recent years, the introduction of deep learning has improved the speed and quality of reconstruction [11,12,13,14,15,33]. Furthermore, if the reconstruction is represented by a neural network, the encoder can also be represented by a neural network. Because coded imaging achieved by shifting the exposure timing can be expressed as a convolution in the time dimension and the reconstruction of video from a captured image can be expressed as deconvolution, we can use deep sensing to jointly optimize the sensing matrix and reconstruction [15,21]. However, a fully controllable complementary metal oxide semiconductor (CMOS) sensor is not realistic owing to hardware limitations [34]. Normal image sensors are designed to capture images in a uniform manner, similar to a global or rolling shutter, and it is difficult for these special sensors to achieve compatibility between sensitivity and resolution because they typically have additional complicated circuits in each pixel, and this decreases the size of the photodiode [35]. Additionally, standard commercial CMOS sensors, e.g., three-transistor CMOS sensors, do not have a per-pixel frame buffer on the chip. Thus, such sensors are incapable of multiple exposures in a non-destructive manner [3]. There is an advanced prototype CMOS sensor [4] that can control the exposure time more flexibly. However, its spatial control is limited to per-line (column and row) operations. Therefore, it is necessary to optimize the exposure patterns by considering the hardware constraints.

In color video sensing, it is necessary to optimize color filters and exposure patterns simultaneously because these encodings are dependent on each other (Figure 1e). Therefore, we propose jointly optimizing the sensing matrix and decoder for compressive color video sensing. The contributions of our paper are summarized as follows:We propose a new pipeline to optimize the color filter, exposure pattern, and reconstruction decoder in compressive color video sensing using a DNN framework [36]. To the best of our knowledge, ours is the first study that considers the actual hardware sensor constraints and jointly optimizes the color filter, exposure patterns, and decoder in an end-to-end manner.The proposed method is a general framework for optimizing the exposure patterns with and without hardware constraints. It is difficult to learn the sensing matrix with a DNN because DNNs can only handle differentiable functions. The optimization of the color filter is an arrangement of the RGB filter, but the exposure pattern is binary and needs to take into account the constraints of the sensor.We demonstrate that the learned exposure pattern can recover high-frame-rate videos with better quality than existing handcrafted and random patterns. Moreover, we demonstrate the effectiveness of our method with images captured by an actual sensor.

This paper is an extended version of the paper [17]. In our ECCV2018 paper [17], we only optimized the exposure pattern, and we used a method of compressing three dimensions into two dimensions. In this paper, the proposed method is a more general method of compressing four dimensions into two dimensions by adding the optimization of color filters and extending it to compressed sensing in space, time, and color.

The other parts of this paper are organized as follows. Section 2 reviews the methods for deep sensing and deep optics, optimization of exposure patterns and reconstruction of video, and optimization of chromatic filter patterns and demosaicing. Section 3 describes the main contribution of this paper: deep sensing for compressive video acquisition under hardware constraints. Section 4 presents the results of the method with simulation and a prototype sensor. Section 5 offers some conclusions.

## 2. Related Studies

### 2.1. Deep Sensing and Deep Optics

In recent years, increasing numbers of studies have been published that implement sensing and reconstruction models using DNNs similar to an encoder–decoder network and jointly optimize the sensing parameters and a reconstruction decoder by end-to-end learning. Such techniques are called deep sensing/optics and have been applied in various domains. Inagaki et al. [18] proposed compressive light-field sensing using a coded aperture camera. They modeled the image acquisition and reconstruction process using a neural network and optimized the aperture patterns as well as a reconstruction decoder with end-to-end training. They were able to reconstruct the 64 × 64 sub-images of the light field from only a few images that were captured with optimized coded aperture patterns. Wu et al. [19] demonstrated a single-frame, single-view, passive 3D imaging technique using a phase mask at the camera aperture. By optimizing the phase mask and reconstruction simultaneously, they designed a phase mask with better performance than related methods. Sun et al. [20] proposed a method for training an autoencoder consisting of an encoder that selects the sampling at the sensor and a decoder for very-long-baseline interferometry (VLBI). The encoder represents the sensor selection as a binary using a fully connected Ising model [37,38]. They demonstrated that their method selects the optimal sensor in the context of a very-long-baseline interferometry array design task. Similar approaches have also been used in compressive video sensing [15,21] and compressive spectral sensing [9,10] tasks, and they provide better performance than previous methods.

### 2.2. Optimization of Exposure Patterns and Reconstruction of Video

Ideal compressive video sensing requires an image captured with a random exposure (Figure 2a), as expressed in Equation (Equation 1) and shown in Figure 1d. Traditionally, random exposure patterns [3,4,8,11,12,33] and model-based reconstruction methods, such as dictionary-based methods [3,4], Gaussian mixture model-based methods [8], the L1 norm minimization problem [5,6,7], and the total variation minimization problem [39], have been used. However, reconstruction in model-based methods with iterative optimization is slow, and with the development of DNNs, DNN-based methods have been mainly used instead. Iliadis et al. [11] proposed a network that is composed of fully connected layers and learns the nonlinear mapping between a video sequence and a captured image. In recent years, researchers have started using convolutional neural network (CNN)-based unrolled networks [12,13,14,15]. Ma et al. [12] proposed a decoding convolutional neural network based on the alternating direction method of multipliers (ADMM). By representing ADMM as a convolutional neural network, they achieved a fast and high-quality reconstruction. Cheng et al. [40] proposed RevSCI-net that is based on multi-group reversible 3D convolutional neural networks. They reduced training time and memory usage by using an end-to-end, multi-loop network. Sun et al. [41] used a residual ensemble network to balance reconstruction time and reconstruction quality. This network aimed to improve reconstruction quality by exploiting spatio–temporal correlations between video frames. Wen et al. [33] applied a transformer base reconstruction network to snapshot compressive video sensing. They solved the problems of CNN-based methods by using a transformer base network to acquire long-range features effectively. Very recently, an end-to-end optimization of the exposure pattern and reconstruction has been proposed because a DNN model performs not only reconstruction but also image capture. Li et al. [15] proposed a network architecture to jointly design the encoding masks and reconstruction method for compressive video sensing. Their network optimizes the full-resolution mask, and they applied Anderson acceleration to the iterative reconstruction network. Martel et al. [42] presented an end-to-end network to jointly optimize a differentiable sensor exposure model and image processing network. By using a function to decode the control signal of the sensor into an exposure pattern, they were able to optimize the shutter function of the sensor with various hardware constraints. However, their sensor model optimizes the exposure control signal for each pixel and is not applicable to sensors with spatial exposure constraints like [4].

Conventional charge-coupled device (CCD) and CMOS sensors either have a global or a rolling shutter. A global shutter exposes all of the pixels concurrently, whereas a rolling shutter exposes every pixel row/column sequentially. A commercial sensor capable of capturing an image with random exposure does not exist. Therefore, most existing studies have only evaluated simulated data [11] or optically emulated implementations [3]. However, Hitomi et al. [3] used an exposure pattern that assumed actual sensor constraints. The sensors that they consider are single-bump exposure (SBE) sensors, which can only control the start time of a single exposure for each pixel and cannot split the exposure duration into multiple exposures in one frame even though the exposure time would be controllable because of the limited exposure control line and the dynamic range, as shown in Figure 2b. There have been only a few attempts to implement sensors for compressive video sensing [34]. Sonoda et al. [4] used a prototype CMOS sensor with exposure control capabilities. The basic structure of this row- and column-wise exposure (RCE) sensor is similar to that of a standard CMOS sensor, although separate reset and transfer signals control the start and end times of the exposure. Because the pixels in a column and row share the reset and transfer signal, respectively, the exposure pattern has row- and column-wise dependency (Figure 2c). These researchers also proposed increasing the randomness of the exposure pattern. However, the method could not completely solve the pattern’s row- and column-wise dependencies.

### 2.3. Optimization of Chromatic Filter Patterns and Demosaicing

A color camera with a single sensor has a chromatic filter array in front of the sensor because a CMOS imager only measures the intensity of the light and cannot discriminate the spectral information. Bayer color filters [1] have been commonly used as color filters. A Bayer filter consists of two green, one blue, and one red pixel in the four-pixel neighborhood, and the pattern uniformly repeats like a checker pattern. This color sensor has only single-channel color information, and the other channels can be interpolated from the neighbor pixels. The color interpolation is usually called demosaicing, and it is well known that demosaicing does not perfectly recover the spatial resolution. Hence, color imaging with a single sensor always sacrifices spatial resolution to obtain colors.

Therefore, a method to obtain color information by compressed sensing using a random CFA was proposed [2,25]. Condat [2] proposed using a random CFA to reduce false colors and the moiré effect. A Bayer filter consists of a checkerboard pattern, and it makes false colors and the moiré effect when capturing scenes with spatial frequencies higher than the sampling frequency. Therefore, by randomizing the arrangement of the CFA, he suppressed the artifacts in scenes containing high spatial frequencies. Moreover, Hirakawa et al. [26] optimized the CFA by analyzing the spatial frequency of the scene. However, the conventional reconstruction method by interpolation also causes artifacts. Therefore, using the compressed sensing theory, Sato et al. [25] reconstructed color images from images acquired with a random CFA using ADMM. These methods deal with encoding and decoding separately, but CFA optimization is a nonlinear and difficult problem. Hence, there have been studies that jointly optimize the color filter and demosaicing model.

Chakrabarti [9] proposed a method that learns optimal color filter patterns jointly with the demosaicing network. They developed an autoencoder that selects a color channel at the encoder and reconstructs it at the decoder. The learned color filters yielded better results than the traditional Bayer filter. There are also studies that, instead of using an RGB filter pattern arrangement, jointly optimize the color filter responses for hyperspectral imaging [10] and material classification [43] from multispectral images.

Since images captured with such color filters have only one channel of information per pixel; it is necessary to complement the missing channel information in order to obtain RGB images. Traditionally, simple interpolation including bicubic interpolation and spline interpolation has been used. However, while simple interpolation is useful when computational resources are limited, it is not suitable for applications that require high-quality images. Therefore, methods for demosaicing by interpolation and refining with CNNs [44,45] and methods for demosaicing using CNNs [46,47,48] have been proposed.

Finally, we summarize the related studies in Table 1.

## 3. Deep Sensing for Compressive Video Acquisition under Hardware Constraints

In this section, we describe the proposed method that jointly optimizes the color filter and exposure pattern for compressive video sensing and then performs reconstruction using a DNN. The proposed DNN consists of two main parts. The first part is the sensing (encoding) layer, which optimizes the arrangement of the color filter and exposure pattern as learnable parameters of a DNN under the constraint imposed by the hardware structure, as described in Section 2.2. The second part is the reconstruction (decoding) layer, which recovers multiple channels and multiple sub-frames from a single captured image that was compressed by using the optimized color filter and exposure pattern. The overall framework is shown in Figure 3.

### 3.1. Sensing Layer

Although color and temporal information are encoded with ϕ in Equation (Equation 1), this process is equivalent to encoding using separate matrices. Due to implementation constraints, the actual color filters and exposure patterns are binary. Therefore, Equation (Equation 1) can be rewritten as follows:(2)$$y\left(w,h\right)=\sum_{t}^{T}\sum_{c}^{C}\phi_{C}\left(w,h,c\right)⊙\left(\phi_{T}\left(w,h,t\right)⊙x\left(w,h,c,t\right)\right),$$
where $\phi_{C}\in {\left[0,1\right]}^{W\times H\times C}$ is a component of the color filter of the sensing matrix, and $\phi_{T}\in {\left[0,1\right]}^{W\times H\times T}$ is a component of the exposure pattern of the sensing matrix. In addition, when considering implementation, color information is encoded by the color filter and temporal information is encoded by the exposure pattern, so they must be encoded separately. Therefore, the sensing layer consists of a color-coding layer and a time-coding layer. The sensing matrix is optimized as learnable parameters of the color filter layer and the exposure pattern layer. Usually, the learnable parameters (weight matrix) ψ of a neural network are continuous values, but the sensing matrix must be binary values. However, binary values cannot be updated because the gradient cannot be calculated. Therefore, the sensing layer has binary parameters for forward and continuous parameters for backward. The continuous parameters are gradually updated by backward, and the binary parameters are updated when the amount of updates is enough to update the binary parameters. This behavior is similar to the sigmoid function commonly used in CNNs, and the gap between binary and continuous parameters has less effect on learning. As the reconstruction layer converges in the later stages of learning, the error propagated to the sensing layer also converges, and the updates to the sensing layer also converge.

Because human vision acquires only the three RGB channels, the camera only needs to acquire RGB channels in many situations. In addition, most of the color filter is placed on the surface of the image sensor for each pixel, and the mechanism for changing it is complicated and impractical. Hence, we use Chakrabarti’s method [9] to optimize the arrangement of a static RGB CFA. Each weight matrix ψ learned by the DNN is usually a continuous value, but a CFA must select one channel. It is possible to select one channel by thresholding the weight matrix ψ of the color filter layer after applying the softmax function to the channel dimension, but the accuracy is not so high. Therefore, we use a softmax function with a parameter β that increases with the number of iterations as follows:(3)$$\phi_{C}\left(i,j,c\right)={\mathrm{Softmax}}_{c}\left[\beta \psi \left(i,j,c\right)\right].$$
where ${\mathrm{Softmax}}_{c}$ is the Softmax function along the *c* dimension. As β increases, the output distribution of the softmax function becomes peaky and finally binary [0,1]. Therefore, we binarize the weight ψ of the CFA while updating it along the gradient by gradually increasing β at each iteration.

We sought an exposure pattern that would be capable of reconstructing video frames with high quality when trained along with the reconstruction (decoding) layer. More importantly, the compressive sensing layer must be capable of handling the exposure pattern constraints imposed by actual hardware architectures. Because the exposure is controlled by the ON/OFF switch on the sensor, the exposure pattern needs to be binary. Further, there are some constraints depending on each sensor due to the arrangement of the signal lines for controlling the exposure, as discussed in the literature ([3,4]).Therefore, we use the method inspired by Martel et al. [42] to optimize the exposure patterns. We consider the following three constraints in this paper.

1.An ideal sensor with a binary exposure pattern but without temporal or spatial constraints (unconstrained sensor);2.A sensor with a binary exposure pattern and with temporal constraints (SBE sensor);3.A sensor with a binary exposure pattern and spatial constraints (RCE sensor).

It is difficult to train a normal neural network with all these constraints. The reason is that the exposure pattern is binary because the exposure is controlled by the electronic shutter, but the weights of the neural network must be continuous values in order to calculate the gradient in the backward propagation. Therefore, the exposure pattern optimization layer learns constrained exposure patterns by changing the weights used for forward and backward propagation. The learning steps for the exposure pattern optimization layer are as follows:1.Initialize the forward weights (binary weights) and the backward weights (continuous weights) with a random pattern that satisfies sensor constraints.2.In the forward propagation, the exposure pattern optimization layer uses the binary weights to simulate a real camera.3.Before backward propagation, the weights of the exposure pattern optimization layer are switched to continuous weights.4.The continuous weights are updated by backward propagation.5.Update the binary weights by using a function that considers the constraints of each sensor on the updated continuous weights.6.Repeat Steps 2 to 5 until the network converges.

In the unconstrained sensor, each pixel can independently control the shutter on/off in each sub-frame. Since the exposure is controlled by the electronic shutter, the exposure pattern is binary. The update function of the binary weight uses a thresholding process because there are no temporal or spatial constraints in the exposure pattern.
(4)$$\psi_{b}\left(i,j,t\right)=\left\{\begin{array}{c} 1\;(\psi_{c}\left(i,j,t\right)\geq Threshold)\\ 0\;(\psi_{c}\left(i,j,t\right)<Threshold) \end{array}\right.$$
where $\psi_{b}\left(i,j,t\right)$ are binary weights of the exposure layer in pixel position (i,j) and sub-frame *t*, and $\psi_{c}\left(i,j,t\right)$ are continuous weights of the exposure layer in pixel position (i,j) and sub-frame *t*. This thresholding function updates the binary weights using continuous weights.

In the SBE sensor, each pixel can control the shutter on/off independently, but this sensor does not have a non-destructive readout and can only expose once in a frame. The dynamic range of the photodiode is limited, so the exposure time must be the same for all pixels. The following function is used for the binary weight update function to consider the temporal constraint.
(5)$$\psi_{b}\left(i,j,t\right)=\left\{\begin{array}{c} 1\;(\tau \leq t<\tau +N)\\ 0\;(other) \end{array}\right.$$
(6)$$\tau =\underset{\tau }{\arg\;\max}\left[\sum_{t=\tau }^{\tau +N-1}\psi_{c}\left(i,j,t\right)\right]$$
where *N* is the number of sub-frames to be exposed. We use Equation (Equation 5) to find the frame with the largest total of the continuous weights and Equation (Equation 6) to update the binary weights.

In the RCE sensor, the signal lines that control the exposure are shared by the vertical or horizontal columns, so the pixels to be exposed are the vertical or horizontal lines. Therefore, the binary weight update function decides whether to expose or not to expose each vertical or horizontal line by convolving the continuous weights for each line. Since there is no time constraint for the RCE sensor, this process is carried out independently for each sub-frame.
(7)$$\psi_{b}\left(i,j,t\right)=\left\{\begin{array}{c} 1\;(\psi_{v}\left(i,t\right)>0\;or\;\psi_{h}\left(j,t\right)>0)\\ 0\;(other) \end{array}\right.$$
(8)$$\psi_{v}\left(i,t\right)=\sum_{j}^{W}\psi_{c}\left(i,j,t\right),\;\psi_{h}\left(j,t\right)=\sum_{i}^{H}\psi_{c}\left(i,j,t\right)$$
where $\psi_{v}\left(i,t\right)$ is the weight corresponding to the vertical exposure control signal, and $\psi_{h}\left(j,t\right)$ is the weight corresponding to the horizontal exposure control signal. We obtain the vertical and horizontal exposure control signals from the continuous weights using Equation (Equation 8) and update the binary weights using Equation (Equation 7) with the exposure control signals.

### 3.2. Reconstruction Layer

We use RevSCI-net [40] as the reconstruction layer. RevSCI-net is composed of three modules: a feature extraction layer $F_{F}$, a feature-level nonlinear mapping layer $F_{R}$, and a reconstruction layer $F_{M}$. The feature extraction layer $F_{F}$ extracts high-dimensional features of the input using 3D CNN layers (the kernel size is 5×5×5, 3×3×3, 1×1×1, and 3×3×3). The input of $F_{F}$ is normalized and expanded to W×H×T using the exposure pattern as follows:(9)$$\bar{y}=y⊘\sum_{t=1}^{T}\phi_{T},\;\;\bar{x}=\bar{y}⊙\phi_{T}$$
where ⊘ is element-wise division, ⊙ is the element-wise product, $\bar{y}\in R^{W\times H}$ is the normalized input, and $\bar{x}\in R^{W\times H\times T}$ is the extended input. The expanded input $\bar{x}$ is converted into 4D features ($R^{W\times H\times C\times T}$) by $F_{F}$.

The nonlinear mapping layer $F_{R}$, which converts the feature extraction layer $F_{F}$ output into video features, uses stacked reversible blocks to reduce memory usage. The original Rev-Net [49] splits the features into two channels, but in RevSCI-net, the authors improved the performance of the video reconstruction task by splitting the features into multiple channels. The ResNet block requires a lot of memory to record all the activations in all the intermediate layers when calculating the gradient by backpropagation, but by dividing the channels in this way, only the last activation needs to be recorded to calculate the gradient, thus saving a lot of memory.

The reconstruction layer $F_{M}$ takes the output of the nonlinear mapping layer $F_{R}$ as input and outputs the reconstructed video. $F_{M}$ is composed of 3D CNN layers (the kernel size is 3×3×3, 3×3×3, 1×1×1, and 3×3×3). Table 2 shows the network architecture of the reconstruction layer.

## 4. Experiments

### 4.1. Experimental and Training Setup

The network was trained by minimizing the errors between the training and reconstructed videos. We used the mean squared error (MSE) as the loss function because it is directly related to the peak signal-to-noise ratio (PSNR):(10)$$L_{\mathrm{MSE}}=\frac{1}{T}\sum_{k=1}^{T}{\left({\hat{x}}_{k}-x_{k}\right)}^{2}$$
where ${\hat{x}}_{k}$ is the $k_{th}$ reconstructed sub-frame and $x_{k}$ is the grand truth of the $k_{th}$ sub-frame. We trained our network using the DAVIS2017 [50] dataset, which is a public benchmarking dataset designed for the task of video object segmentation. We used 6864 videos with rotation and flipping for training. We used Adam as an optimizer and set the learning rate of the decoder to 0.0002 and the batch size to 8, and we trained for 20 epochs. Experiments were run on a computer with an Intel Xeon Silver 4116 processor running at 2.10 MHz, using 100 GB of RAM and an NVIDIA Quadro GV100, and running Windows 10 Education.

### 4.2. Simulation Experiments

We carried out color compressive video sensing simulation experiments to evaluate joint optimization. We compared the proposed method with GMM [8], Multi-Layer Perception (MLP) [17], ADMM-Net [12], and RevSCI-net [40] to evaluate its performance. GMM [8] represents video as GMM and reconstructs videos by estimating the conditional expectation value. MLP [17] is a method we proposed at ECCV2018 that jointly optimizes the reconstruction layer (MLP), the color filter, and the exposure pattern. The MLP consists of four hidden layers, and ReLU truncates each layer. Additionally, MLP reconstructs the video in 8×8 pixel patches, and the final output video is the concatenation of the patches. ADMM-Net [12] uses a neural network for denoising in the alternating direction method of multipliers (ADMM). These methods [8,12,17] target monochrome videos, but we modified these methods to be able to apply them to color videos. RevSCI-net [40] uses a Bayer filter [1] and a random exposure pattern and trains only the reconstruction layer. We assumed three different hardware constraints for the SBE [3] (Figure 2b), RCE [4] (Figure 2c), and unconstrained sensors (Figure 2a). The SBE sensor has a temporal constraint in the exposure pattern, and the RCE sensor has a spatial constraint in the exposure pattern. The details of the SBE and RCE sensor constraints are described in Section 2.2. The exposure pattern for an unconstrained sensor can independently control the exposure for each pixel and achieve perfect random exposure, which is ideal in compressive sensing theory. Each parameter of the handcrafted pattern for the unconstrained sensors was randomly chosen. The network size was determined based on the size of the patch volume to be reconstructed. We used a sensor with controllable exposure [4], which exposes an 8×8 pixel block. For color video, the volume size of W×H×C×T was set to 256×256×3×16 in the experiments.

We generated a captured image simulated for the random and optimized color filter using the SBE, RCE, and unconstrained sensors. We input the simulated images to the reconstruction network to recover the video, and we quantitatively evaluated the reconstruction quality using the PSNR and structural similarity (SSIM). PSNR and SSIM were calculated in RGB components. We used nine 256×256 pixel videos with 16 sub-frames in the evaluation. Figure 4 shows two example results from the Aerobatics and Chameleon videos. The upper half of Figure 4 (Aerobatics) shows that GMM and MLP have blurred images due to block noise, and the lips are not sharp in ADMM-Net and RevSCI-net. On the other hand, the proposed method reconstructs the lips sharply. In the bottom half of Figure 4 (Chameleon), the chameleon’s movement is too large to be reconstructed as blurred in GMM and MLP. In addition, GMM has artifacts caused by Bayer filters. In RevSCI-net, the object is less blurred than in MLP, but it is not sharp. ADMM-Net and the proposed method reconstruct the eye of the chameleon sharply. The reconstructed video by ADMM-Net is noisy in the moving parts, and the reconstructed video by the proposed method is too smooth. Table 3 shows the average PSNRs and SSIMs of GMM [8], MLP [17], ADMM-Net [12], RevSCI-net [40], and the proposed method for the SBE, RCE, and unconstrained sensors. The SSIMs of the results of the proposed method are best for all types of hardware constraints, and the PSNRs of the proposed method are better than the other methods for the SBE and RCE sensors. In the unconstrained sensor, there is little difference in reconstruction quality between the proposed method and ADMM-Net, as the randomly sensing matrices can compress enough information, and the large search space makes convergence difficult. RevSCI-net has better reconstruction quality than GMM and MLP, but the proposed method can further improve the reconstruction quality by optimizing the color filter and exposure pattern.

We analyzed the effectiveness of the proposed method by calculating p-values for the PSNRs of the proposed and previous methods. The *p*-values in RevSCI-net are $6.4\times 10^{-3}$ for the SBE sensor, $2.0\times 10^{-4}$ for the RCE sensor, and $6.0\times 10^{-5}$ for the unconstrained sensor, and they are less than 0.05 for all sensor constraints. This means that the joint optimization of the sensing matrix and reconstruction improves the reconstruction quality. Similarly, the *p*-values of GMM (the SBE sensor: $1.7\times 10^{-3}$, the RCE sensor: $4.6\times 10^{-4}$, and the unconstrained sensor: $3.3\times 10^{-4}$) and MLP (the SBE sensor: $1.9\times 10^{-5}$, the RCE sensor: $8.1\times 10^{-5}$, and the unconstrained sensor: $6.1\times 10^{-5}$) are less than 0.05, showing that the proposed method is better. On the other hand, in ADMM, the *p*-values (the SBE sensor: 0.40, the RCE sensor: 0.40, and the unconstrained sensor: 0.46) are larger than 0.05, and the proposed method is not better in reconstruction quality. However, by applying the proposed joint optimization method to ADMM, the reconstruction quality of ADMM can be improved.

GMM, a traditional iterative optimization method, is very time-consuming to reconstruct. On the other hand, the reconstruction time for one video is less than one second for reconstruction using a neural network. The proposed method is about 1.5 times faster than ADMM-Net, and there is little difference between the proposed method and RevSCI-net because they use the same reconstruction model.

We show a box plot of the PSNR reconstructed with GMM, MLP, ADMM-Net, RevSCI-net, and the proposed method for each sensor constraint in Figure 5. As shown in Figure 5, the proposed method has better average quality and smaller variance than the other methods in each sensor constraint.

To quantitatively verify the contributions of the optimization of the color filter and the exposure pattern, we compared the reconstruction quality with and without optimization of the color filter and the exposure pattern (shown in Table 4). We compared four networks, such as with and without color filter optimization and with and without exposure pattern optimization, for three sensor constraints. The reconstruction quality was better for all sensor constraints when both the color filter and the exposure pattern were optimized. The reconstruction quality was also better when either the color filter or the exposure pattern was optimized than when neither was optimized. Figure 6 shows the percentage of pixels that changed in the exposure state for each epoch. We also show the exposure pattern for each epoch. The figure shows that the exposure pattern converges in the first few epochs, and the exposure state of most pixels does not change after the 10th epoch. Figure 7 compares random and optimized sensing matrices. In a Bayer filter (Figure 7a), each color is arranged alternately. On the other hand, each color is continuously arranged by optimization in the optimized filter (Figure 7b). In the case of a random exposure pattern (Figure 7c), the exposure is divided into small time slices, but with optimization (Figure 7d), the exposures are more continuous. Similar results have been reported by [21], even though our study considered the hardware constraints in pattern optimization. These results support the advantage of deep sensing, which jointly optimizes the sensing matrix and the reconstruction network.

Our method optimizes the color filters and the exposure patterns for the training data; i.e., it can generate the color filters and the exposure patterns specialized for a particular context. We used the Karlsruhe Institute of Technology and Toyota Technological Institute (KITTI) dataset [51] to evaluate the effect of optimizing for a special dataset. Optimizing the color filter, exposure pattern, and decoder for a specific dataset can improve reconstruction quality compared to optimizing in general. For comparison, we used the random color filter and the random exposure pattern (same as RevSCI-net [40]) and the generally optimized color filter and exposure pattern. We used eighty scenes for training and four scenes for testing. We cropped the images in the dataset and reshaped them to 256×256. Figure 8 shows the results of reconstruction, and Table 5 shows the average PSNRs and SSIMs of each exposure pattern. The centerline in Figure 8 is reproduced as optimized for KITTI, but the edge of the centerline in the generally optimized one is not clear. In the random color filter and the random exposure pattern, the separation of the centerline cannot be seen. In this way, optimization for special context provides better reconstruction quality than optimization in general.

### 4.3. Experiments with the Prototype Sensor

We conducted experiments using an actual compressive image captured by the camera with the sensor reported by [4,52] to demonstrate that the optimized exposure pattern can be implemented in an actual image sensor. Figure 9 shows the camera image used in the experiment. This camera was made by Hamamatsu Photonics K.K. This sensor has no color filter, so we optimized only the exposure pattern in this experiment. The pixel size of the prototype sensor is 7.4 × 7.4 μm^2^. The number of total pixels is 672×512, and the number of effective pixels is 656×496. The size of the light-receiving area is 4.8544×3.6704 mm, and the dynamic range is 60 dB. The exposure is controlled by a field-programmable gate array (FPGA). The compressed video was captured at 71 frames per second (fps). We employed 16 exposure patterns per frame. Thus, the reconstructed video was equivalent to 1136 fps after recovering all of the 16 sub-frames. We used the exposure pattern obtained by the sensing layer of the proposed network after the training. Moreover, we reconstructed the video from the captured image using the reconstruction layer of the proposed network. The sensor had a rolling shutter readout and temporal exposure patterns, which were temporally shifted according to the position of the image’s row. The exposure pattern was shifted every 32 rows (four blocks with an 8×8 patch) in the case where the resolution of the sensor was 672×512 pixels and the number of exposure patterns was 16 in one frame. For example, the actual sub-exposure pattern was applied to the first four blocks as sub-exposure patterns 0–15, the second four blocks were applied as patterns 1–15 and 0, and the third four blocks were applied as patterns 2–15, 0, and 1, and so on. Hence, we trained 16 different reconstruction networks to apply various shifted exposure patterns. We used these different reconstruction networks every 32 rows in an image. Since our prototype sensor is not capable of dynamically updating exposure patterns, we fixed the sensing matrix optimized by the neural network when capturing the actual scene with the prototype sensor. Figure 10 shows a single frame of the actual captured image and three of the 16 reconstructed sub-frames. The upper row shows that dropping whitewater appears at different positions in the reconstructed sub-frames, and the motion and shape have been recovered. The bottom row shows a blinking eye. Our method successfully recovered very different appearances. Because the scenes are substantially different from the videos included in the training dataset, these results also demonstrate the generalization ability of the trained network.

Since time information is encoded, the dynamic range of the sensor is restricted as a limitation of the coded exposure. In addition, if each pixel has a different exposure time, the dynamic range is more restricted because the pixel with the longest exposure time must be captured so that it is not saturated. If some pixels are saturated, the temporal information cannot be reconstructed correctly. However, image sensors often have more bits (the sensor we used had 14 bits), whereas a standard image has 8 bits, and temporal information can be convolved into this gap.

## 5. Conclusions

In this paper, we first argued that real sensor architectures for developing controllable exposure have various hardware constraints that make the implementation of compressive video sensing based on completely random exposure patterns impractical. To address this issue, we proposed a general framework that consists of sensing and reconstruction layers using a DNN. Additionally, we jointly optimized the encoding and decoding models under the hardware constraints. Our proposed method is applicable to various neural-network-based video reconstruction methods and to image sensors with various control constraints. This makes the proposed method applicable to other reconstruction methods and image sensors to be proposed in the future.

We presented examples of applying the proposed framework to sensors with different constraints, i.e., the SBE, RCE, and unconstrained sensors. We demonstrated that our optimal patterns and decoding network reconstructed videos with higher quality than the videos reconstructed with the handcrafted patterns in simulation and real experiments.

Experimental results show that joint optimization improves the reconstruction quality even when the same reconstruction method is used.

In future work, we will apply the proposed method to applications other than video reconstruction. An example of another application is Time-of-Fright (ToF) imaging. Since ToF requires very high temporal resolution in the image sensor, the distance resolution can be improved by applying our proposed method.

## Figures and Tables

**Figure 1 sensors-23-07535-f001:**
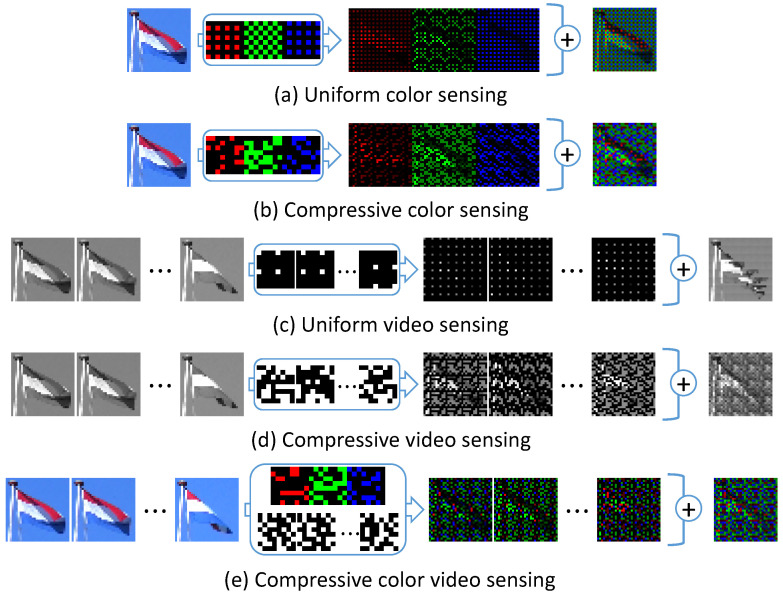
Capturing images with a sensing matrix. The captured image is the sum of the element-wise product between the scene and the sensing matrix. (**a**) Uniform color image sensing with the Bayer filter. The spatial resolution of each color is decreased to 1/c. (**b**) Compressive color sensing with a non-uniform filter. Non-uniform observations make it easy to reconstruct the original signal. (**c**) Uniform video sensing with a uniform exposure pattern. Neighboring pixels are considered one large pixel, and each sub-pixel is exposed at a different time, reducing the spatial resolution but increasing the temporal resolution. (**d**) Compressive video sensing with a non-uniform exposure pattern. Spatio–temporal information is convolved into a captured image with non-uniform sampling. (**e**) Compressive color video sensing with a non-uniform color filter and exposure pattern. Color and spatio–temporal information are convolved into a two-dimensional image.

**Figure 2 sensors-23-07535-f002:**
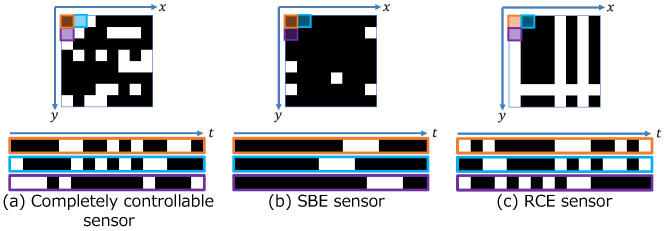
Examples of exposure patterns under hardware constraints. (**a**) Completely controllable sensor: it can be freely sampled in spatio–temporal dimensions. (**b**) SBE sensor [3]: it has temporal constraints on sampling. (**c**) RCE sensor [4]: it has spatial constraints on sampling.

**Figure 3 sensors-23-07535-f003:**
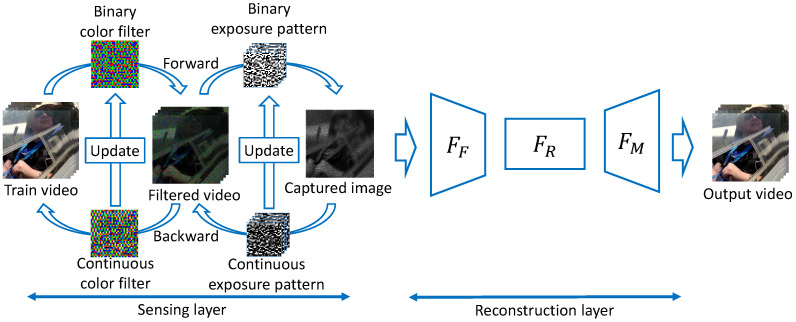
Network Structure. Proposed network structure to jointly optimize the color filter and exposure pattern of compressive video sensing and perform reconstruction using a DNN. The left side represents the sensing layer that compresses video into an image using the color filter and exposure pattern. The right side represents the reconstruction layer that learns the nonlinear mapping between the compressed image and full-color video for reconstruction. ($F_{F}$: a feature extraction layer. $F_{R}$: a feature-level nonlinear mapping layer. $F_{M}$: a reconstruction layer).

**Figure 4 sensors-23-07535-f004:**
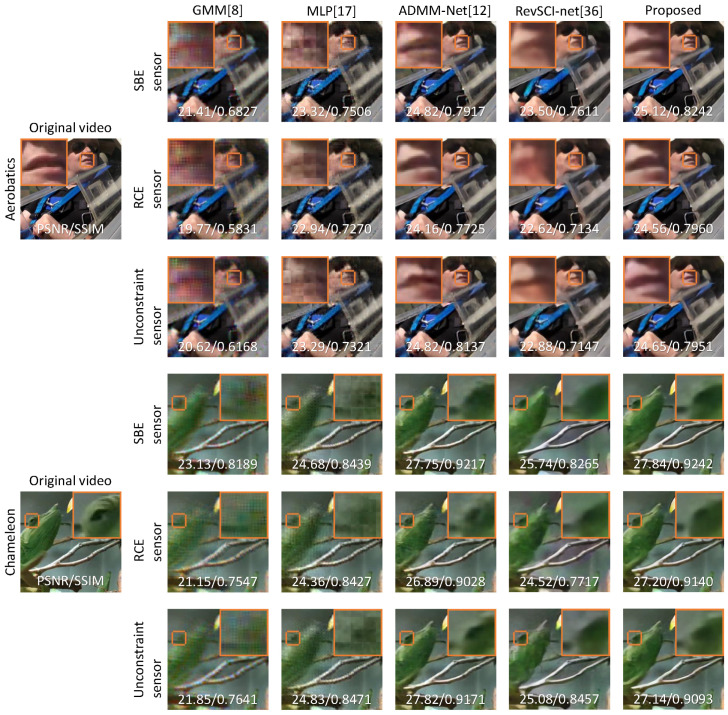
Reconstruction results. We show the results reconstructed by GMM [8], MLP [17], ADMM-Net [12], RevSCI-net [40], and the proposed method with three sensor constraints: SBE, RCE, and unconstrained. Scenes from Aerobatics of the 10th sub-frame (top) and Chameleon of the 10th sub-frame (bottom). The top left or right corners of the images show an enlarged region of the images.

**Figure 5 sensors-23-07535-f005:**
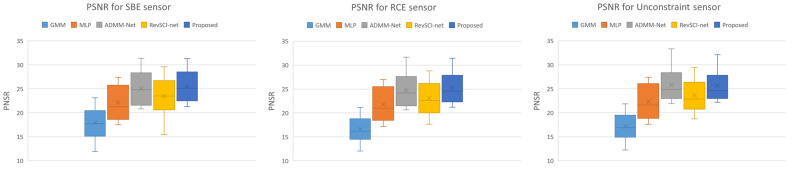
Box plot of PSNR reconstructed with GMM, MLP, ADMM-Net, RevSCI-net, and the proposed method. The proposed method has the best quality and the smallest variance.

**Figure 6 sensors-23-07535-f006:**
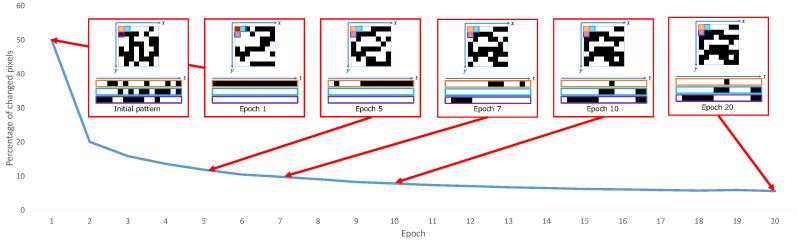
Percentage of changed pixels at each epoch. We also show the exposure patterns at each epoch. The exposure pattern converges in the first few epochs, and only 10% or fewer of the pixels change after the 10th epoch.

**Figure 7 sensors-23-07535-f007:**
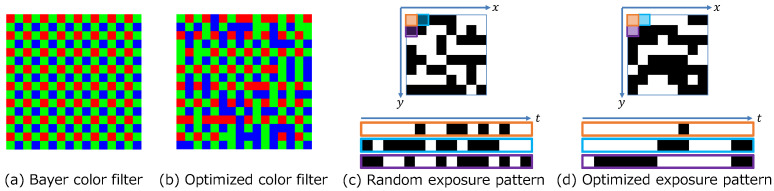
Comparison of random and optimized sensing matrices. We show the top left part of 16×16 pixels in color filters and 8×8 pixels in exposure patterns for the unconstrained sensor.

**Figure 8 sensors-23-07535-f008:**
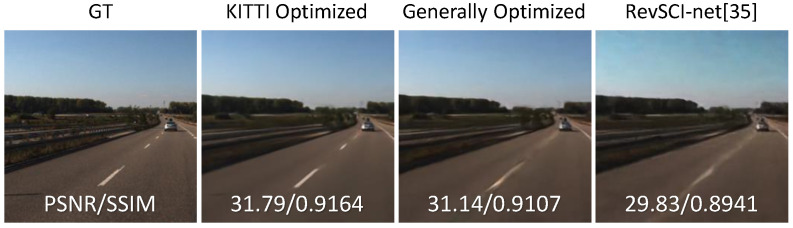
Reconstruction results of the 5th sub-frame in KITTI dataset. Reconstruction quality can be improved by tuning the color filter and exposure pattern to the specific task.

**Figure 9 sensors-23-07535-f009:**
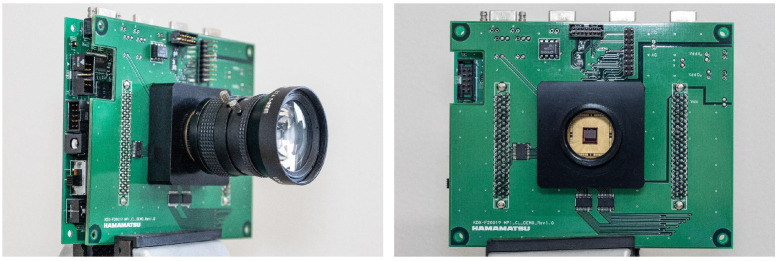
Appearance of the prototype camera made by Hamamatsu Photonics K.K. We used this prototype camera to capture the actual scene.

**Figure 10 sensors-23-07535-f010:**
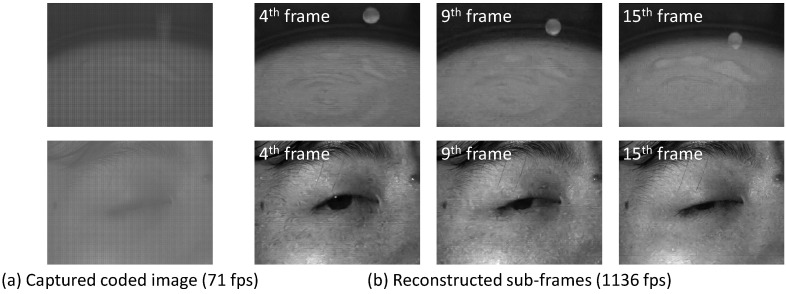
Reconstruction results in the actual scenes. We reconstructed 16 frames of video from the captured image: (**a**) shows the captured image, and (**b**) shows the 4th, 9th, and 15th frames of the reconstructed video. Upper row: dropping white water. Bottom row: blinking eye.

**Table 1 sensors-23-07535-t001:** Summarization of the related studies.

	Application	Sensing Matrix	Method
Inagaki et al. [18]	Compressive light-field sensing	Joint optimization	Neural network
Wu et al. [19]	Passive 3D imaging		
Sun et al. [20]	VLBI		
Hitomi et al. [3]	Compressive video sensing	Random	Model-based
Sonoda et al. [4]			
Liu et al. [5]			
Azghsni et al. [6]			
Zhao et al. [7]			
Yang et al. [8]			
Yuan et al. [39]			
Ma et al. [12]	Compressive video sensing	Random	Neural network
Wen et al. [33]			
Cheng et al. [40]			
Sun et al. [41]			
Iliadis et al. [21]	Compressive video sensing	Joint optimization	Neural network
Li et al. [15]			
Martel et al. [42]			
Condat [2]	RGB imaging	Random	Model-based
Sato et al. [25]			
Hirakawa et al. [26]	RGB imaging	Optimization	Model-based
Tan et al. [44]	RGB imaging	Random	Neural network
Tan et al. [45]			
Gharbi et al. [46]			
Kokkinos et al. [47]			
Park et al. [48]			
Chakrabarti [9]	RGB imaging	Joint optimization	Neural network
Nie et al. [10]	Compressive hyperspectral sensing	Joint optimization	Neural network
Saragadam et al. [43]			

**Table 2 sensors-23-07535-t002:** Network architecture of reconstruction layer.

Module	Operation	Kernel	Stride	Output Size
$F_{F}$	Conv.	5×5×5	(1, 1, 1)	16×T×H×W
	Conv.	3×3×3	(1, 1, 1)	16×T×H×W
	Conv.	1×1×1	(1, 1, 1)	32×T×H×W
	Conv.	1×3×3	(1, 2, 2)	64×T×H/2×W/2
ResNet block in $F_{R}$	Conv.	3×3×3	(1, 1, 1)	32×T×H/2×W/2
	Conv.	3×3×3	(1, 1, 1)	64×T×H/2×W/2
$F_{M}$	Conv. Transpose	1×3×3	(1, 2, 2)	32×T×H×W
	Conv.	3×3×3	(1, 1, 1)	16×T×H×W
	Conv.	1×1×1	(1, 1, 1)	16×T×H×W
	Conv.	3×3×3	(1, 1, 1)	C×T×H×W

**Table 3 sensors-23-07535-t003:** Average PSNR/SSIM of nine color video reconstructions and average reconstruction times.

		GMM [8]	MLP [17]	ADMM-Net [12]	RevSCI-Net [40]	Proposed
SBE sensor	PSNR	17.91	22.13	25.04	23.48	**25.50**
	SSIM	0.6531	0.7185	0.8109	0.6996	**0.8300**
RCE sensor	PSNR	16.58	21.84	24.76	23.10	**25.23**
	SSIM	0.5637	0.7086	0.7822	0.7030	**0.8109**
Unconstrained	PSNR	17.16	22.32	**25.89**	23.64	25.69
sensor	SSIM	0.5756	0.7118	0.8102	0.7343	**0.8253**
	Time	162.3 s	0.2050 s	0.02109 s	0.01378 s	0.01411 s

**Table 4 sensors-23-07535-t004:** Average PSNR/SSIM with and without optimization of the color filter and the exposure pattern.

		Handcrafted	Optimized	Handcrafted	Optimized	Random	Optimized
		**SBE**	**SBE**	**RCE**	**RCE**	**Unconstrained**	**Unconstrained**
Random	PSNR	23.48	24.20	23.10	23.50	23.63	24.52
filter	SSIM	0.6996	0.7443	0.7030	0.6913	0.7343	0.7776
Optimized	PSNR	23.71	25.50	25.02	25.23	24.32	25.69
filter	SSIM	0.7820	0.8300	0.8046	0.8109	0.7687	0.8253

**Table 5 sensors-23-07535-t005:** Average PSNR/SSIM of reconstructed video in KITTI dataset.

	KITTI Optimized	Generally Optimized	RevSCI-net [40]
PSNR	27.53	27.01	25.57
SSIM	0.8594	0.8507	0.8220

## Data Availability

Not applicable.

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
