# Peer review of "Deep Sensing for Compressive Video Acquisition†"

_sensors, 2023, doi:10.3390/s23177535_

Round 1

Reviewer 1 Report

The paper is presented very clearly and I recommend the publication of this paper as it is because the objective is very clear in the abstract and explained with details in the discussion section.

Author Response

We thank you for your careful check and for accepting our contributions.

Please refer to the attached file for comments to other reviewers.

Reviewer 2 Report

This study argued that real sensor architectures for developing controllable exposure have various hardware constraints that make the implementation of compressive video sensing based on completely random exposure patterns impractical. To address this issue, this research proposed a general framework that consists of sensing and reconstruction layers using a DNN and jointly optimized the encoding and decoding models under the hardware constraints. Finally, apply the proposed framework to sensors with different constraints, i.e., the SBE, RCE, and unconstrained sensors.

1. In the experiment, two videos were used for testing. From the results, the difference is not large. Whether the quality of the video will affect the results is not discussed in the article. In addition, the system conversion time is not provided, so it is difficult to judge whether the overall performance is better.

2. The PSNR/SSIM is not clear in Tables 2, 3.

3. Figure 10, the definitions are not clear about the 4th, 9th, and 15th in the reconstructed sub-frames.

4. The characteristics of the system architecture superior to other systems and the reasons for the design of the system can be discussed in detail in the conclusion.

Author Response

We thank you for your careful check and for accepting our contributions.

We detail our responses to your comments below.

Please refer to the attached file for comments to other reviewers.

>1. In the experiment, two videos were used for testing. From the results, the difference is not large. Whether the quality of the video will affect the results is not discussed in the article. In addition, the system conversion time is not provided, so it is difficult to judge whether the overall performance is better.

Feedback:
We used nine videos for our experiments, two of which are shown in Fig 4.
The advantage of the proposed method can be verified by comparing RevSCI-net with the proposed method. The upper half of Fig.4 (Aerobatics) shows that the proposed method reconstructs the lips sharper than RevSCI-net. The key of the proposed method is joint optimization of exposure pattern and reconstruction, and the reconstruction quality is improved by joint optimization. Although there is little difference in the reconstruction quality compared to ADMM-Net, we expect the reconstruction quality to be improved by applying joint optimization to ADMM-Net. And we added reconstruction time in Table 3.

>2. The PSNR/SSIM is not clear in Tables 2, 3.

Feedback:
We rewrote Table 3,4 to make PSNR and SSIM easier to understand. (The table numbers are shifted because we added a table.)

>3. Figure 10, the definitions are not clear about the 4th, 9th, and 15th in the reconstructed sub-frames.

Feedback:
We reconstruct 16 frames of video from a single captured image. We showed the 4th, 9th, and 15th frames of the reconstructed video in Fig 10. We rewrote the caption in Fig 10.

>4. The characteristics of the system architecture superior to other systems and the reasons for the design of the system can be discussed in detail in the conclusion.

Feedback:
We added a statement to the conclusion section about the advantage of joint optimization of sensing matrix and reconstruction.
"Our proposed method is applicable to various neural network-based video reconstruction methods and to image sensors with various control constraints.
This makes the proposed method applicable to other reconstruction methods and image sensors to be proposed in the future."
(Line 479)

Reviewer 3 Report

This paper proposes an end-to-end learning approach that jointly optimizes the sampling pattern as well as the reconstruction decoder. The authors applied this deep sensing approach to the video compressive sensing problem. They modeled the spatio–temporal sampling and color filter pattern using a convolutional neural network constrained by hardware limitations during network training. They demonstrated that the proposed method performed better than the manually designed method in gray-scale video and color video acquisitions.

It is a valuable study. The manuscript is well-organized and well-written. References are sufficient and appropriate.

The followings can be fixed:

1- What about running time (execution time) of the method?

2- Providing a table that summarizes the related work would increase the understandability of the difference from the previous studies in the "Related Works" section. 

3- The authors may explain the possible future studies in the conclusion section.

4- A concern is that no formal statistical analysis of the results are done, to indicate whether the differences in performance are statistically significant or not.

For example; Friedman Aligned Rank Test, Wilcoxon Test, Quade Test, etc.  

p-value can be calculated and compared with the significance level (p-value < 0.05). 

5- It would be useful if the conclusion would be supported by the results.

6- The organization of the paper (the structure of the manuscript) may be written at the end of the "Introduction" section. 

For example: "Section 2 presents ... Section 3 gives ...." 

7- In line 29, DNN appears for the first time, and the full name of the word should be placed here.

Some abbreviations are used in the text without giving their expansion.   

For example; SSIM, KITTI, CMOS, etc. 

The authors should write that "these abbreviations stand for what".

-

Author Response

We thank you for your careful check and for accepting our contributions.

We detail our responses to your comments below.

Please refer to the attached file for comments to other reviewers.

>1- What about running time (execution time) of the method?

Feedback:
We added reconstruction time in Table 3.

>2- Providing a table that summarizes the related work would increase the understandability of the difference from the previous studies in the "Related Works" section.

Feedback:
Thanks for the helpful advice. We added a table summarizing related studies at the end of Section 2.

>3- The authors may explain the possible future studies in the conclusion section.

Feedback:
We thank the reviewer for a reasonable suggestion. We mentioned the other application of the proposed method in the conclusion section.
"In future work, we will apply the proposed method to applications other than video reconstruction. An example of another application is Time of Fright (ToF) imaging. Since the ToF requires a very high temporal resolution in the image sensor, the distance resolution can be improved by applying our proposed method."
(Line 487)

>4- A concern is that no formal statistical analysis of the results are done, to indicate whether the differences in performance are statistically significant or not. For example; Friedman Aligned Rank Test, Wilcoxon Test, Quade Test, etc.   p-value can be calculated and compared with the significance level (p-value < 0.05).

Feedback:
We analyzed the effectiveness of the proposed method by calculating p-values for the PSNR of the proposed and previous methods. The p-values for GMM, MLP, and RevSCI-net are smaller than 0.05, suggesting that the proposed method has better reconstruction quality. On the other hand, in ADMM, the p-value is larger than 0.05, and the proposed method is not better in reconstruction quality. However, by applying the proposed joint optimization method to ADMM, the reconstruction quality of ADMM can be improved.
We added an analysis of p-value to the paper. (Line 389)

>5- It would be useful if the conclusion would be supported by the results.

Feedback:
The key idea of our proposed method is joint optimization. Our proposed method is effective because the reconstruction quality is improved compared to RevSCI-net, which has the same reconstruction layer as our proposed method. We added a statement to our conclusions about the advantage of joint optimization.
"Our proposed method is applicable to various neural network-based video reconstruction methods and to image sensors with various control constraints.
This makes the proposed method applicable to other reconstruction methods and image sensors to be proposed in the future."
(Line 479)
"Experimental results show that joint optimization improves the reconstruction quality even when the same reconstruction method is used."
(Line 486)

>6- The organization of the paper (the structure of the manuscript) may be written at the end of the "Introduction" section. 
For example: "Section 2 presents ... Section 3 gives ...."

Feedback:
We thank the reviewer for a reasonable suggestion. We added the organization of the paper at the end of the introduction.

>7- In line 29, DNN appears for the first time, and the full name of the word should be placed here. Some abbreviations are used in the text without giving their expansion. For example; SSIM, KITTI, CMOS, etc. The authors should write that "these abbreviations stand for what".

Feedback:
W
e thank the reviewer for a careful check. We added explanations for all abbreviations.